# A Detailed Study of Electronic and Dynamic Properties of Noble Gas–Oxygen Molecule Adducts

**DOI:** 10.3390/molecules27217409

**Published:** 2022-11-01

**Authors:** Caio Vinícius Sousa Costa, Guilherme Carlos Carvalho de Jesus, Luiz Guilherme Machado de Macedo, Fernando Pirani, Ricardo Gargano

**Affiliations:** 1Instituto de Física, Universidade de Brasília, Brasília 70297-400, DF, Brazil; 2Universidade Federal de São João del Rei, Divinópolis 35501-296, MG, Brazil; 3Dipartimento di Chimica, Biologia e Biotecnologie, Universitá degli studi di Perugia, Via Elce di Sotto 8, 06123 Perugia, Italy; 4Istituto CNR di Scienze e Tecnologie Chimiche (CNR-SCITEC), Via Elce di Sotto 8, 06123 Perugia, Italy

**Keywords:** noble gas–O_2_ adducts, spectroscopic constant, lifetime, charge transfer, energy decomposition

## Abstract

In this work, the binding features of adducts formed by a noble gas (Ng = He, Ne, Ar, Kr, Xe, and Rn) atom and the oxygen molecule (O_2_) in its ground Σg−3, in the past target of several experimental studies, have been characterized under different theoretical points of view to clarify fundamental aspects of the intermolecular bond. For the most stable configuration of all Ng–O_2_ systems, binding energy has been calculated at the theory’s CCSD(T)/aug-cc-pVTZ level and compared with the experimental findings. Rovibrational energies, spectroscopic constants, and lifetime as a function of temperature were also evaluated by adopting properly formulated potential energy curves. The nature of the interaction involved was deeply investigated using charge displacement analysis, symmetry-adapted perturbation theory (SAPT), and natural bond orbital (NBO) methods. In all adducts, it was found that the charge transfer plays a minor role, although O_2_ is an open shell species exhibiting a positive electron affinity. Obtained results also indicate that the dispersion attraction contribution is the main responsible for the complex stability.

## 1. Introduction

Molecular interactions are essential in various areas of fundamental and applied research. The growing demand for new technologies has driven the study of weakly bound or long-range molecular complexes, controlled by non-covalent interactions, whose ubiquitous components are the van der Waals ones. Species with closed electronic shells, such as noble gas atoms, can form weakly bound stoichiometric aggregates (van der Waals) in the high-pressure regime. Several research groups have focused on understanding the stability and nature of the interatomic interactions involved in these complexes. Many phenomena have been observed in its dense metallic phases such as the appearance of electronic levels in the band gaps. This has been possible thanks to doping with atomic impurities that favor changes in electronic properties at low pressures [1]. In its ground electronic state Σg−3, O_2_ is an open-shell paramagnetic molecule with a positive electron affinity and its interaction in different phases with other partners is of great relevance. In its singlet ground state, the gaseous dimer O_2_–O_2_ has been proposed by Lewis [2] as a prototype of the weak chemical bond (see also V. Aquilanti et al. [3]). Moreover, in the solid state, O_2_ forms three phases with different ranges of stability and magnetic character (see V. Aquilanti et al. [3] and references therein). Experimental studies reveal that changes in crystallography and resistivity of platinum thin films (deposited by sputtering at increasing O_2_ partial pressures) are enhanced by the use of Ne as a gas carrier. A slower deposition rate on Ne may allow more time for oxide formation in the substrate [4]. High-pressure experiments of the binary phase diagrams of O_2_ with He, Ne, Ar, and Xe noble gas at 296 K have been performed. The knowledge of these binary phase diagrams is very important as it provides a reference dataset to test theoretical calculations on mixtures, allows the growth of a single crystal of O_2_ in a medium of the pressure of Ng, and also synthesizes oxides of Ng at high pressure. Furthermore, thermodynamic and structural properties of the mixtures of O_2_ with Ng have been studied widely [5]. From a theoretical point of view, it is important and challenging to understand whether a van der Waals compound exists in any mixture of O_2_ with Ng, as already experimentally observed for Ar(H_2_)_2_ [6], Xe(H_2_)_7_ [1], and He(N_2_)_11_ [7]. This knowledge can help in the synthesis of new molecular materials by pressure [5].

On the other hand, molecular interactions play a crucial role in different areas of knowledge. The growing demand for new technologies has driven the study of weakly bound (non-covalent) aggregates. In fact, the detailed characterization of the main component involved in a non-covalent interaction becomes fundamental in the identification and modeling of possible terms that compose the long-range or van der Waals-type forces [8,9]. This objective is very useful for evaluating the dynamic and static properties of these types of aggregates under a wide variety of possible applications. Gaseous Ng–O_2_ systems have been the target of several investigations, exploiting essentially molecular beam scattering experiments which provided important details on range, strength, and anisotropy of the interaction (see F. Pirani et al. [10] and references therein). Moreover, in some particular cases, as Ar–O_2_, some spectroscopic features of the IR spectrum (see, for instance, G. Henderson and G. Ewing [11]) have been also resolved. For Ne–O_2_, the Zeeman spectrum has been measured [12]. In general, the characterization of important spectroscopic features and of the balance of the leading interaction components, determining the intermolecular bond strength and range in Ng–O_2_, is still not completely available for the complete family of these weakly bound systems. In addition, several experimental findings suggest that Ng–O_2_ represents a prototype of anisotropic van der Waals interaction, but a theoretical confirmation of this finding is still lacking, due to the difficulty to evaluate weak interactions in systems involving open-shell species.

Based on these reports, the present work presents a broad study involving the oxygen O_2_ and the noble gases—Ng (Ng = He, Ne, Ar, Kr, Xe, and Rn). In more detail, exploring a series of different methodologies, the potential energy curves of the Ng–O_2_ complexes, the charge displacement, the decomposition of electronic energy, the rovibrational energies, the spectroscopic constants, and the lifetime (as a function of temperature) were evaluated. With the determination of these important properties, it was possible to investigate the role and nature of the weak interaction in such compounds. The present investigation represents a continuation of a previous study on the structure and reactivity of noble gas compounds [13].

## 2. Methodologies and Computational Details

All the geometric variables used to describe the anisotropic interactions in Ng–O_2_ systems are defined according to Figure 1. In this study, the interaction in the most stable configuration and the isotropic-spherical potential are both represented by the well-known Improved Lennard–Jones (ILJ) function [14] that provides in analytical form their radial dependence, that is, the dependence of involved interaction potential on the distance R between Ng atom and the center of mass of O_2_.

For the case of a complex formed by two neutral species as Ng and O_2_, the ILJ function is given by:(1)$$V\left(R\right)=D_{e}\left[\frac{6}{n(R)-6}{\left(\frac{R_{e}}{R}\right)}^{n(R)}-\frac{n(R)}{n(R)-6}{\left(\frac{R_{e}}{R}\right)}^{6}\right].$$

In the above equation, n(R) is expressed by $\beta +4{\left(\frac{R}{R_{e}}\right)}^{2}$ and the β parameter reports the softness/hardness of the constituents that make up the adducts. Experimentally, this parameter assumes the value 9 for compounds involving noble gases [14]. Basically, to use Equation (Equation 1), the equilibrium distance ($R_{e}$) and the dissociation energy ($D_{e}$) must be known. $R_{e}$ and $D_{e}$, associated with absolute minimum configurations of the complete family of Ng–O_2_ compounds were determined by varying R distance and φ angle (between 0 and 180°) and keeping the intramolecular distance between oxygen atoms in O_2_ fixed at the equilibrium position ($R_{o-o}=1.208$ [15]), as represented in Figure 1. For each generated configuration, it was calculated the energy solving the electronic equation (within the Born–Oppenheimer approximation) at CCSD(T) [16,17] /aug-cc-pVTZ [18] level for He–O_2_, Ne–O_2_, Ar–O_2_, and Kr–O_2_ adducts. For the Xe–O_2_ and Rn–O_2_ compounds, the electronic energy was calculated at CCSD(T)/aug-cc-pVTZ-PP level. Furthermore, the basis set superposition error correction [19,20] was taken into account for all studied complexes. All calculations were performed via the Gaussian09 computational code [21]. Obtained *D_e_* and *R_e_* potential features are consistent with isotropic potential parameters predicted by empirical correlation formulas (ECF) [22] which provide the basic potential features for non-covalent interaction exploiting exclusively fundamental physical properties of interacting partners such as the electronic polarizability. In turn, predicted results agree with the experimental values extracted from the analysis of quantum interference effects resolved in scattering experiments and directly probing basic features of the interaction between projectile and target (see V. Aquilanti et al. [23] and reference therein). Therefore, present theoretical results have been used to perform an extensive and internally consistent analysis of the nature of the intermolecular interaction (see below). They have been exploited, together with results of ECF and Equation (Equation 1), to evaluate, in an internally consistent way, basic spectroscopic features for the complete family of systems.

The rovibrational spectroscopic constants ($\omega_{e}$, $\omega_{e}x_{e}$, $\omega_{e}y_{e}$, $\alpha_{e}$, and $\gamma_{e}$) were obtained using two different procedures. The first was the Dunham method [24], which is determined through the derivatives of potential energy curves in the equilibrium configuration. The second is given by the following equations [25]:(2)$$\begin{array}{l} \omega_{e}=\frac{1}{24}\left[141\left(E_{1,0}-E_{0,0}\right)-93\left(E_{2,0}-E_{0,0}\right)+23\left(E_{3,0}-E_{1,0}\right)\right]\\ \omega_{e}x_{e}=\frac{1}{4}\left[13\left(E_{1,0}-E_{0,0}\right)-11\left(E_{2,0}-E_{0,0}\right)+3\left(E_{3,0}-E_{1,0}\right)\right]\\ \omega_{e}y_{e}=\frac{1}{6}\left[3\left(E_{1,0}-E_{0,0}\right)-3\left(E_{2,0}-E_{0,0}\right)+\left(E_{3,0}-E_{1,0}\right)\right]\\ \alpha_{e}=\frac{1}{8}\left[-12\left(E_{1,1}-E_{0,1}\right)+4\left(E_{2,1}-E_{0,1}\right)+4\omega_{e}-23\omega_{e}y_{e}\right]\\ \gamma_{e}=\frac{1}{4}\left[-2\left(E_{1,1}-E_{0,1}\right)+\left(E_{2,1}-E_{0,1}\right)+2\omega_{e}x_{e}-9\omega_{e}y_{e}\right]. \end{array}$$

In Equation (Equation 2), $E_{v,j}$ expresses the rovibrational energy where *v* and *j* represent the vibrational and rotational quantum numbers, respectively. In this work, $E_{v,j}$ were calculated by solving the nuclear Schrödinger equation using the Discrete Variable Representation method [26].

The description of the nature of the intermolecular bond of O_2_–Ng adducts was based on the definition of charge displacement along a direction (*z*-axis, for example) and is defined by the following equation [27,28,29,30]:(3)$$\Delta q\left(z\right)=\int_{-\infty }^{\infty }dx\int_{-\infty }^{\infty }dy\int_{-\infty }^{z}\Delta \rho \left(x,y,z^{\prime }\right)dz^{\prime },$$
where Δq is the density difference between the compound and the two separated parts (O_2_ and noble gas) arranged in the same positions they occupy in the compound. We emphasize that canonical charge decomposition methods cannot be used since they provide inaccurate results for Δq when the charge displacement is small [31]. The *z*-axis that appears in Equation (Equation 3) joins the bonding center of the O_2_ molecule (located at the origin of the *z*-axis) with the noble gas (located in the negative part of the *z*-axis). Accordingly, Δq determines at each *z*-position the electron charge that is moved from the right to the left side along the negative *z*-axis. In this way, the displacement of charge takes place from the Ng to the oxygen dimer when Δq(z) is negative. If Δq(z) is positive, then the electron charge occurs from O_2_ to Ng. Equation (Equation 3) was solved using the Multiwfn computational package [32].

With the aim of individualizing the contribution of the terms (Electrostatic $E_{elect}$, induction $E_{ind}$, dispersion $E_{disp}$, and exchange $E_{exch}$) that compose the interaction involved in the Ng–O_2_ complexes, the symmetry-adapted perturbation theory (SAPT) [33] method was used at sapt2+3(CCD)/aug-cc pVTZ [18] level as implemented in the PSI4 [34,35] computational code. In order to describe the origin of the electronic rearrangements of the Ng–O_2_ complexes in detail, the analysis of the natural bond orbital (NBO) was also used [36]. All NBO calculations were performed at CCSD(T) [16,17]/aug-cc-pVTZ [18] level by using the Gaussian09 package [21]. For Xe–O_2_ and Rn–O_2_ complexes, aug-cc-pVTZ-PP were employed.

Finally, the lifetimes as a function of the temperature of all Ng–O_2_ complexes were determined using Slater’s theory and it is given by expression [37,38]:(4)$$\tau \left(T\right)=\frac{1}{\omega_{e}}e^{\frac{D_{e}-E_{0,0}}{R_{g}T}},$$
where $E_{0,0}$ stands for zero point energy (the rovibrational energy calculated for v=0 and j=0), *T* is the temperature, and $R_{g}$ is the universal gas constant. This description assumes that the unimolecular decomposition of the aggregate occurs when the interaction coordinate arrives at the dissociation threshold (*D_e_*).

## 3. Results and Discussion

Table 1 shows the CCSD/aug-cc-pVDZ optimized geometric parameters (*R_e_* and φ) and the CCSD(t)/aug-cc-pVTZ corresponding energy (*D_e_*) that describe the most stable configuration of all atom–molecule complexes formed by Ng and O_2_. The single reference coupled cluster calculation for the closed shell system is considered reliable if the T1 diagnostic value is below 0.020 [39,40]. The T1 values for all systems investigated show that there is no multireference character, and it decreases from He to Rn as follows: He–O_2_ (0.0141), Ne–O_2_ (0.0128), Ar–O_2_ (0.0119), Kr–O_2_ (0.0099), Xe–O_2_ (0.0099), and Rn–O_2_ (0.0091). These results were compared with those obtained via the ECF procedure [9,22]. ECF makes use of a generalized connection between the potential parameters involved in a van der Waals interaction, with polarizabilities and the number of valence electrons (from the fragments of the complexes) that are effectively perturbed by the interaction. Comparing the two results, it is noted that the greatest (smallest) difference found for *R_e_* and *D_e_* was 0.28 Å (0.08 Å) for He–O_2_ (Xe–O_2_), and 1.62 meV ≈ 0.037 kcal/mol (0.00 meV) for Rn–O_2_ (Xe–O_2_), respectively. This comparison indicates that there is a good agreement between the *R_e_* e *D_e_* results obtained by the two methodologies. This fact indicates that the potential energy curves (PEC), for each complex, constructed from the substitution of the values of β=9.0, *R_e_* and *D_e_* (Table 1) in Equation (Equation 1) are suitable for describing the electronic and dynamic properties of Ng–O_2_ compounds. The corresponding PECs are shown in Figure 2 and Figure 3. Another essential fact that deserves to be highlighted is the excellent agreement between the experimental ($\varphi =90^{\circ }$) and theoretical angles of the most stable approach of He ($\varphi =89.9^{\circ }$) and Ar ($\varphi =88.8^{\circ }$) to the binding center of the O_2_ molecule ([10] and references therein and [41]).

Using the PECs from Figure 2 and Figure 3, the reduced masses from Table 1, and the procedure described in Section 2, it was possible to determine the rovibrational energies of the Ng–O_2_ complexes as presented in Table 2 and Table 3. From these tables, it is important to note that both He–O_2_ PECs admit only one confined vibrational level within its potential well. This happens probably due to the small values of the dissociation energy (*D_e_*) and reduced mass of the He–O_2_ compound. The number of levels obtained with the different types of potential energy curves is the same, except for Ne–O_2_, Ar–O_2_, and Rn–O_2_, where the potential well of PEC ECF contains one more level than CSDD(T) PEC. Table 4 and Table 5 show the rovibrational spectroscopic constants obtained by both the Dunham method and Equation (Equation 2). From these tables, one can see a good agreement between the two methodologies. This agreement is important because it brings more confidence in the obtained results since, as far as we know, there are no literature data for comparison. Note that for the compounds He–O_2_ (which has only one vibrational level inside its potential well) and Ne–O_2_ (which has only three vibrational levels inside its potential well), there was no possibility to calculate the spectroscopic constants via Equation (Equation 2), because to use it, at least four vibrational levels are needed.

Figure 4 shows the calculated charge displacement (Δq) for all Ng–O_2_ adducts in their most stable configuration. From this Figure, one can see that Δq becomes appreciably different from zero only when Ng is very close to O_2_. However, in all cases, Δq changes sign within the intermolecular distance, and only for Ar–O_2_, it maintains a very small negative value in an appreciable z range. All these features represent a clear indication that the charge transfer plays a practically null role in determining the weak intermolecular bond [27,28,29,30].

Table 6 shows the interaction energy decomposition of the Ng–O_2_ complexes calculated at sapt2+3(CCD)/aug-cc-pVTZ level. From this table, it is possible to observe that the dispersion term ($E_{disp}$) overcomes in all Ng–O_2_ adducts, with a higher/lower contribution of 82.2%/75.6% for the Ar–O_2_/Ar–O_2_ complexes. This fact suggests that all Ng–O_2_ complexes are basically governed by a non-covalent or van der Waals-type interaction, where exchange repulsion and dispersion attraction represent the leading components. Table 7 reports the second-order perturbation energies (E^2^) obtained through the NBO analysis for the Ng–O_2_ complexes. First, the results presented in this table reveal that the electronic donation between the Ne atom and the O_2_ oxygen dimer is practically negligible with E^2^ less than 0.05 kcal/mol. In addition, a very small electronic donation takes place from the oxygen 1-center valence lone pair orbital (LP) to the 1-center antibond orbital of Rydberg (RY*) of the He, Kr, Xe, and Rn noble gases. For the Ar–O_2_, an small electronic donation also happens from the O–O bond orbital (BD), located on O_2_ dimer, to the 1-center antibond orbital of Rydberg (RY*) of the Ar atom. In any case, these values are very small and consistent with the charge displacement results of Figure 3. This combined analysis confirms the nature of van der Waals for all these systems.

Figure 5 and Figure 6 present the lifetime as a function of temperature for all complexes obtained through CCSD(T) and ECF PECs, respectively. The first indication of these figures is that the He–O_2_ complex has a lifetime of less than one second for the entire temperature range from 200 to 500 K. In this case, according to Wolfgang [42], the potential energy well is not deep enough to exclude the intermediate complex and so the adduct is considered unstable. For the other complexes, the lifetime was slightly above 1 picosecond for the entire considered temperature range, indicating that these systems are weakly bound. These facts are in line with the results obtained with the charge displacement, NBO analysis, and SAPT calculations.

## 4. Conclusions

The present theoretical investigation proves that Ng–O_2_ aggregates are effectively bound by van der Waals interactions. Although O_2_ in its ground electronic state (Σg−3) is an open-shell species with positive electron affinity, the present analysis confirms the experimental finding that its interaction with Ng atoms is not affected by charge transfer component, even in adducts formed by O_2_ with heavier Ng. It is also confirmed that in all systems the T (perpendicular) is the most stable configuration. The phenomenological potentials, which correctly reproduce interference effects in scattering experiments that are depending on specific features of the potential well, are here used, together with ab initio calculations of the intermolecular interaction, to evaluate roto-vibration spectroscopic features of Ng–O_2_ complexes in the range of temperatures 200–500 K. It is found that only He–O_2_ is unstable under the selected conditions of the bulk. The knowledge acquired in the present study about the non-covalent character (van der Waals) of the mixture of O_2_ and Noble gas can be useful in the synthesis of new molecular materials by pressure.

## Figures and Tables

**Figure 1 molecules-27-07409-f001:**
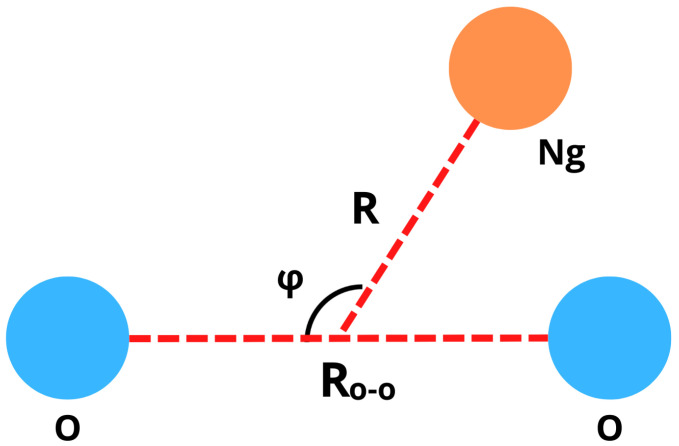
Geometric variables defining the structure of Ng–O_2_ compounds.

**Figure 2 molecules-27-07409-f002:**
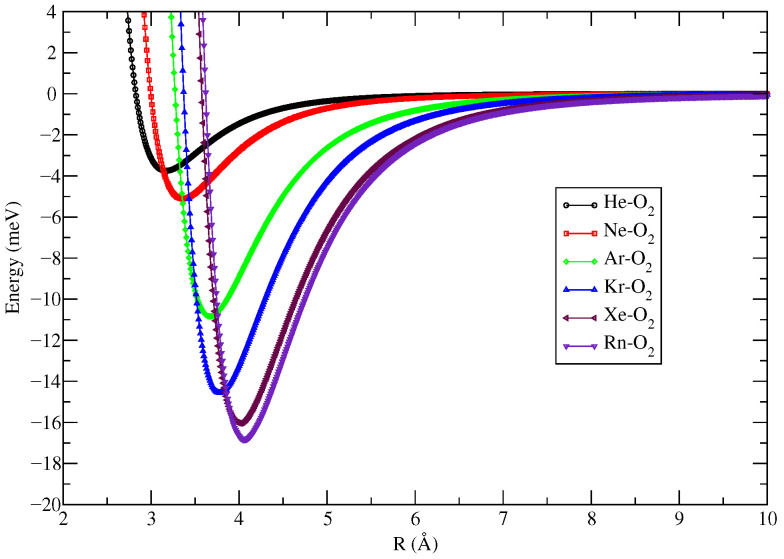
Improved Lennard Jones potential energy curves for the Ng–O_2_ compounds obtained at CCSD(t)/aug-cc-pVTZ level with the BSSE correction.

**Figure 3 molecules-27-07409-f003:**
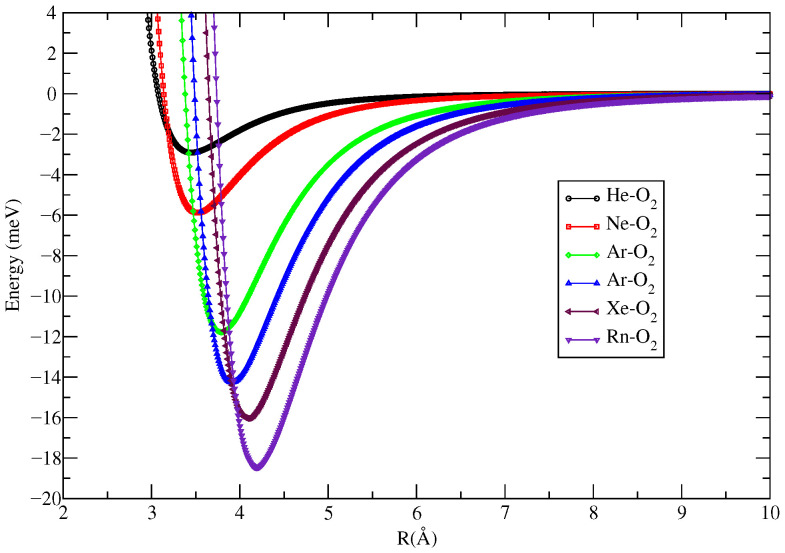
Improved Lennard Jones potential energy curves for the Ng–O_2_ compounds obtained through the ECF approach.

**Figure 4 molecules-27-07409-f004:**
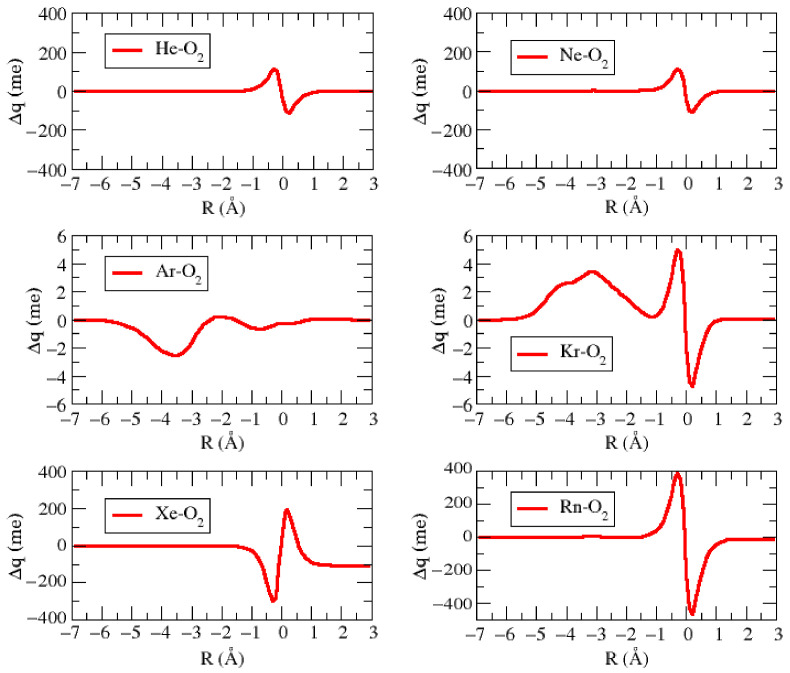
Charge Displacement (Δq) pictures for the most stable configuration for the Ng–O_2_ complexes.

**Figure 5 molecules-27-07409-f005:**
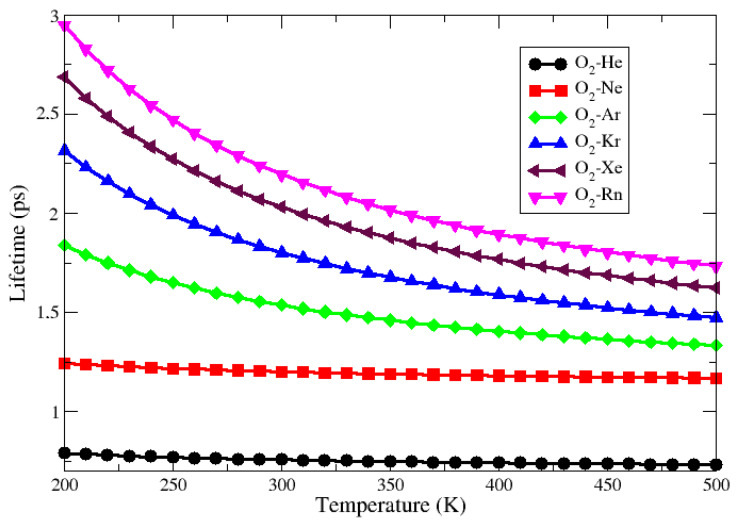
Lifetime behavior as a function of temperature for a Ng–O_2_ compounds obtained at CCSD(t)/aug-cc-pVTZ level.

**Figure 6 molecules-27-07409-f006:**
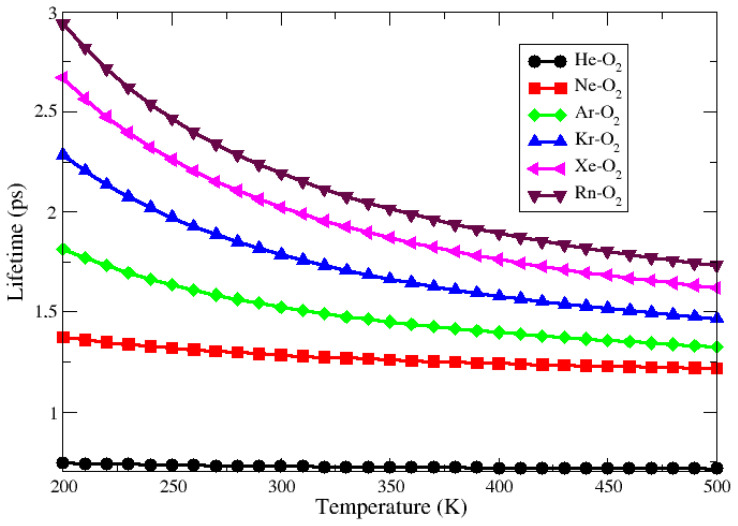
Lifetime behavior as a function of temperature for the Ng–O_2_ compounds obtained through ECF approach.

**Table 1 molecules-27-07409-t001:** Geometric parameters (*R_e_* and φ) of the most stable configurations and corresponding energies (*D_e_*) of the Ng–O_2_ complexes. The values in parentheses refer to the results available in the literature which were determined via the ECF approach [9,22] (see text for more details). The μ reduced masses of each complex are also shown.

Complexes	*R_e_* (Å)	*D_e_* (meV)	*φ* (°)	*μ* (a.u)
He–O_2_	3.17 (3.45)	3.75 (2.91)	89.9	6309.80486
Ne–O_2_	3.36 (3.52)	5.11 (5.88)	80.0	22,558.61103
Ar–O_2_	3.67 (3.79)	10.84 (11.78)	88.8	32,374.47312
Kr–O_2_	3.78 (3.91)	14.56 (14.26)	81.0	42,210.71200
Xe–O_2_	4.01 (4.09)	16.04 (16.04)	81.0	46,896.81884
Rn–O_2_	4.06 (4.19)	16.84 (18.46)	81.0	52,658.65576

**Table 2 molecules-27-07409-t002:** Vibrational (j=0) and rovibrationl (j=1) energies in cm^−1^ for the Ng–O_2_ complexes obtained through CCSD(T)/aug-cc-pVTZ potential energy curve.

*v*	*j*	He–O_2_	Ne–O_2_	Ar–O_2_	Kr–O_2_	Xe–O_2_	Rn–O_2_
0	0	21.3536	13.2550	14.7518	14.7332	13.9043	13.3169
1	0	-	30.9952	39.6642	40.8599	39.0358	37.6220
2	0	-	39.2232	58.7146	62.6703	60.6929	58.8989
3	0	-	-	72.2600	80.3115	78.9669	77.2163
4	0	-	-	80.8756	93.9888	93.9772	92.6617
5	0	-	-	85.4532	103.9947	105.8846	105.3487
6	0	-	-	-	110.7416	114.9070	115.4270
7	0	-	-	-	114.7757	121.3358	123.0943
8	0	-	-	-	116.8906	125.5402	128.6035
9	0	-	-	-	-	128.0208	132.2635
10	0	-	-	-	-	-	134.5124
0	1	22.0513	13.4736	14.8871	14.8320	13.9837	13.3861
1	1	-	31.1662	39.7873	40.9526	39.1111	37.6879
2	1	-	39.3342	58.8238	62.7560	60.7636	58.9612
3	1	-	-	72.3532	80.3898	79.0326	77.2748
4	1	-	-	80.9502	94.0586	94.0376	92.7161
5	1	-	-	85.5065	104.0551	105.9391	105.3986
6	1	-	-	-	110.7916	114.9551	115.4721
7	1	-	-	-	114.8142	121.3767	123.1341
8	1	-	-	-	116.9200	125.5735	128.6377
9	1	-	-	-	-	128.0470	132.2914
10	1	-	-	-	-	-	134.5351

**Table 3 molecules-27-07409-t003:** Vibrational (j=0) and rovibrational (j=1) energies in cm^−1^ for the Ng–O_2_ complexes obtained through the ECF potential energy curve.

*ν*	*j*	He–O_2_	Ne–O_2_	Ar–O_2_	Kr–O_2_	Xe–O_2_	Rn–O_2_
0	0	15.3999	13.1298	14.9448	14.1078	13.6404	13.5385
1	0	-	32.3170	40.5221	39.2010	38.3479	38.4275
2	0	-	42.9573	60.5738	60.2558	59.7129	60.4665
3	0	-	46.9871	75.3899	77.4049	77.8210	79.7135
4	0	-	-	85.4212	90.8305	92.7828	96.2402
5	0	-	-	91.3598	100.7913	104.7460	110.1375
6	0	-	-	94.2410	107.6498	113.9102	121.5227
7	0	-	-	-	111.8883	120.5411	130.5474
8	0	-	-	-	114.2040	124.9771	137.4060
9	0	-	-	-	-	127.6664	142.3384
10	0	-	-	-	-	-	145.6317
11	0	-	-	-	-	-	147.8315
0	1	16.0302	13.3341	15.0720	14.2003	13.7171	13.6037
1	1	-	32.4855	40.6387	39.2878	38.4204	38.4897
2	1	-	43.0807	60.6786	60.3364	59.7811	60.5256
3	1	-	47.0579	75.4811	77.4785	77.8846	79.7694
4	1	-	-	85.4970	90.8966	92.8413	96.2926
5	1	-	-	91.4178	100.8489	104.7990	110.1862
6	1	-	-	94.2820	107.6981	113.9572	121.5672
7	1	-	-	-	111.9263	120.5815	130.5876
8	1	-	-	-	114.2332	125.0104	137.4415
9	1	-	-	-	-	127.6928	142.3687
10	1	-	-	-	-	-	145.6567
11	1	-	-	-	-	-	147.8534

**Table 4 molecules-27-07409-t004:** Ng–O_2_ spectroscopic constants (cm^−1^) obtained through the CCSD(T)/aug-cc-pVTZ potential energy curve.

Constants	He–O_2_	Ne–O_2_	Ar–O_2_	Kr–O_2_	Xe–O_2_	Rn–O_2_
$\omega_{e}$ (Equation (Equation 2))	-	-	31.11	30.58	28.69	27.40
$\omega_{e}$ (Dunham)	47.81	29.69	31.00	30.56	28.67	27.38
$\omega_{e}x_{e}$ (Equation (Equation 2))	-	-	3.19	2.27	1.80	1.56
$\omega_{e}x_{e}$ (Dunham)	21.90	5.32	3.10	2.23	1.79	1.55
$\omega_{e}y_{e}$ (Equation (Equation 2))	-	-	$5.95\times 10^{-2}$	$2.46\times 10^{-2}$	$1.50\times 10^{-3}$	$1.15\times 10^{-2}$
$\omega_{e}y_{e}$ (Dunham)	0.84	$8.46\times 10^{-2}$	$2.76\times 10^{-2}$	$1.46\times 10^{-2}$	$1.00\times 10^{-2}$	$7.90\times 10^{-3}$
$\alpha_{e}$ (Equation (Equation 2))	-	-	$5.31\times 10^{-3}$	$2.87\times 10^{-3}$	$1.96\times 10^{-3}$	$1.56\times 10^{-3}$
$\alpha_{e}$ (Dunham)	0.16	$1.68\times 10^{-2}$	$5.49\times 10^{-3}$	$2.91\times 10^{-3}$	$1.98\times 10^{-3}$	$1.56\times 10^{-3}$
$\gamma_{e}$ (Equation (Equation 2))	-	-	$4.08\times 10^{-4}$	$1.42\times 10^{-4}$	$8.13\times 10^{-5}$	$5.55\times 10^{-5}$
$\gamma_{e}$ (Dunham)	$3.72\times 10^{-2}$	$1.48\times 10^{-3}$	$2.72\times 10^{-4}$	$1.05\times 10^{-4}$	$6.15\times 10^{-5}$	$4.42\times 10^{-5}$

**Table 5 molecules-27-07409-t005:** Ng–O_2_ spectroscopic constants (cm^−1^) obtained through the ECF potential energy curve.

Constants	He–O_2_	Ne-O_2_	Ar–O_2_	Kr–O_2_	Xe–O_2_	Rn–O_2_
$\omega_{e}$ (Equation (Equation 2))	-	29.59	31.38	29.26	28.13	27.79
$\omega_{e}$ (Dunham)	38.70	28.52	31.29	29.23	28.11	27.78
$\omega_{e}x_{e}$ (Equation (Equation 2))	-	5.72	2.98	2.12	1.73	1.47
$\omega_{e}x_{e}$ (Dunham)	18.54	4.84	2.90	2.09	1.72	1.46
$\omega_{e}y_{e}$ (Equation (Equation 2))	-	0.32	$4.83\times 10^{-2}$	$2.21\times 10^{-2}$	$1.42\times 10^{-2}$	$9.67\times 10^{-3}$
$\omega_{e}y_{e}$ (Dunham)	0.74	7.31	$2.40\times 10^{-3}$	$1.33\times 10^{-2}$	$9.42\times 10^{-3}$	$6.86\times 10^{-3}$
$\alpha_{e}$ (Equation (Equation 2))	-	$1.32\times 10^{-2}$	$4.65\times 10^{-3}$	$2.62\times 10^{-3}$	$1.85\times 10^{-3}$	$1.35\times 10^{-3}$
$\alpha_{e}$ (Dunham)	0.14	$1.45\times 10^{-2}$	$4.78\times 10^{-3}$	$2.66\times 10^{-3}$	$1.87\times 10^{-3}$	$1.36\times 10^{-3}$
$\gamma_{e}$ (Equation (Equation 2))	-	$2.36\times 10^{-3}$	$3.21\times 10^{-4}$	$1.26\times 10^{-4}$	$7.26\times 10^{-5}$	$4.38\times 10^{-5}$
$\gamma_{e}$ (Dunham)	$3.41\times 10^{-2}$	$1.22\times 10^{-3}$	$2.20\times 10^{-3}$	$1.49\times 10^{-5}$	$5.68\times 10^{-5}$	$3.55\times 10^{-5}$

**Table 6 molecules-27-07409-t006:** Interaction energy decomposition (given in Kcal/mol) of the Ng–O_2_ complexes calculated at the sapt2+3(CCD)/aug-cc-pVTZ level.

Terms	He–O_2_	Ne–O_2_	Ar–O_2_	Kr–O_2_	Xe–O_2_	Rn–O_2_
$E_{elect}$	−0.0062	−0.035	−0.1141	−0.1571	−0.0776	−0.1022
$E_{exch}$	0.0342	0.1449	0.3539	0.4578	0.2543	0.3224
$E_{ind}$	−0.0014	−0.0024	−0.015	−0.0229	−0.0176	−0.0231
$E_{disp}$	−0.0900	−0.2457	−0.5407	−0.6381	−0.4416	−0.4977
${\%E}_{elect}$	13.8%	13.7%	17.0%	19.2%	13.6%	16.4%
${\%E}_{ind}$	10.6%	9.5%	2.3%	2.8%	3.2%	3.7%
${\%E}_{disp}$	75.6%	76.8%	80.7%	78.0%	82.2%	79.9%

**Table 7 molecules-27-07409-t007:** Natural bond orbital population analysis obtained through second-order perturbation energies (E^2^) at the CCSD(T)/aug-cc-pVTZ level for the Ng–O_2_ complexes.

Complexes	Donor	Receptor	E^2^(kcal/mol)
He–O_2_	LP(2) O2	RY*(1) He	0.06
Ne–O_2_	-	-	-
Ar–O_2_	BD(2) O1-O2	RY*(1) Ar	0.12
Kr-O_2_	LP(2) O1 LP(2) O2	RY*(1) Kr RY*(1) Kr	0.05 0.06
Xe–O_2_	LP(2) O1 LP(2) O2	RY*(1) Xe RY*(1) Xe	0.06 0.06
Rn–O_2_	LP(2) O1 LP(2) O2	RY*(2) Rn RY*(1) Rn	0.06 0.06

## Data Availability

Not applicable.

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
