# Peer review of "A Detailed Study of Electronic and Dynamic Properties of Noble Gas–Oxygen Molecule Adducts"

_molecules, 2022, doi:10.3390/molecules27217409_

Round 1

Reviewer 1 Report

In this manuscript, Costa, Gargano, and co-workers investigated the binding features and spectroscopic of Ng-O2 with a range of distinct methods. The authors confirm that dispersion interactions are the main responsible for the stability of the complexes and that charge transfer does not play a significant role despite dioxygen’s positive electron affinity in its triplet ground state. I believe the paper is of interest to both experimentalists and theoreticians working on noble gas compounds and I am happy to recommend the acceptance of this work, provided that the authors successfully address the following points.   

1 - The authors mention that for determining the Re of Ng-O2, the O-O bond was fixed at the equilibrium position. Did the authors try to relax the O-O bond and perform a full geometry optimization? How important is the O-O bond relaxation energy? 

2 - Is there any multireference character in the computed Ng-O2 systems? The authors should present T1 diagnostics for their CCSD(T) calculations. If multireference character is indeed present, the authors could also consider performing additional calculations using more appropriate coupled-cluster flavors, such as the left-eigenstate completely renormalized coupled-cluster, CRCC(2,3).

3 - Grammar corrections

- In the abstract, line 7, consider changing "life" for "lifetime".

- The basis set for systems bearing heavy Ng atoms is sometimes mentioned as aug-cc-pVTZ-PP or aug-cc-pVTZ-Pseudo-Potentials. Please choose only one nomenclature.

- Line 161: T "is" the temperature.

- Table 2 caption: "(j = 1)" must appear before "energies".

Author Response

Professor Dennis Zhou

Assistant Editor of Molecules

After performing a careful and thorough review of our manuscript we managed to reply and argue about each question that the referees pointed out. We are very grateful for the high quality of the review reports which provided us with a great opportunity to considerably improve our paper. We have assigned each of the following sections to a referee, and performed the modifications according to the suggestions of the referee, as follows:

Reviewer 1

In this manuscript, Costa, Gargano, and co-workers investigated the binding features and spectroscopic of Ng-O2 with a range of distinct methods. The authors confirm that dispersion interactions are the main responsible for the stability of the complexes and that charge transfer does not play a significant role despite dioxygen’s positive electron affinity in its triplet ground state. I believe the paper is of interest to both experimentalists and theoreticians working on noble gas compounds and I am happy to recommend the acceptance of this work, provided that the authors successfully address the following points. 

1) Referee’s comment: The authors mention that for determining the Re of Ng-O2, the O-O bond was fixed at the equilibrium position. Did the authors try to relax the O-O bond and perform a full geometry optimization? How important is the O-O bond relaxation energy?

Authors’ answer:  We performed some optimization tests considering the O-O bond relaxed. The obtained results were similar to those performed considering the O-O bond fixed. We believe that this similarity is due to the fact that the Ng-O2 interaction potentials generate a weak intermolecular bond, whose dissociation energy falls in the 3-18 meV (0.07-0.42 Kcal/mol) energy scale, and exhibit a limited anisotropic character.

2) Referee’s comment: Is there any multireference character in the computed Ng-O2 systems? The authors should present T1 diagnostics for their CCSD(T) calculations. If a multireference character is indeed present, the authors could also consider performing additional calculations using more appropriate coupled-cluster flavors, such as the left-eigenstate completely renormalized coupled-cluster, CRCC(2,3).

Authors’ answer:  The single reference coupled cluster calculation for the closed-shell system is considered reliable if the T1 diagnostic value is below 0.020 ( See T. J. Lee, P. R. Taylor, International Journal of Quantum Chemistry, v.36, p.199-207, 1989 and T. J. Lee, Chemical Physics Letters, v.372, p.362-367, 2003). The T1 values for all systems investigated to show that there is no multireference character, and it decreases from He to Rn as follows: He-O2 (0.0141), Ne-O2 (0.0128), Ar-O2 (0.0119), Kr-O2 (0.0099), Xe-O2 (0.0099) and Rn-O2 (0.0091). This text was added in the first paragraph of Section 3 (Results and Discussions).

3) Grammar corrections

  1. a) Referee’s comment: In the abstract, line 7, consider changing "life" to "lifetime".

Authors’ answer:  In the abstract (line 7), the word “life” was replaced by “lifetime”.

  1. b) Referee’s comment: The basis set for systems bearing heavy Ng atoms is sometimes mentioned as aug-cc-pVTZ-PP or aug-cc-pVTZ-Pseudo-Potentials. Please choose only one nomenclature.

Authors’ answer:  In the new version of the manuscript, we used only the aug-cc-pVTZ-PP nomenclature.

  1. c) Referee’s comment: Line 161: T "is" the temperature.

Authors’ answer:  As suggested by Referee, this correction was performed.

  1. d) Referee’s comment: Table 2 caption: "(j = 1)" must appear before "energies"

Authors’ answer: As suggested by Referee, "(j = 1)" is before "energies".

Reviewer 2 Report

Weak Interactions, including van der Waals forces, are tremendously important and help in the development of novel molecular materials. In this work, the authors have presented a thorough computational study to understand the non-covalent interactions between molecular oxygen and noble gases. The computational data are further correlated with experimental results. In order to understand the existence of weak interactions, binding energies between noble gas-oxygen (Ng-O2) adducts were calculated. Furthermore, charge displacement analysis, symmetry-adapted perturbation theory (SAPT), and natural bond orbital (NBO) analysis were performed. Based on the computations and experiments, it was concluded that the Ng-O2 aggregates are effectively bound by van der Waals interactions and the interactions are not affected by the charge transfer between the two. The study is interesting and will attract the attention of a broad range of audiences interested in the development of novel materials using weak forces. The work has been competently performed with a high scientific standard, and the results have been scholarly presented. Though the manuscript reads well, the introduction, results and discussion are divided into several small paragraphs which should be compiled into long paragraphs including the conclusions. Therefore, I recommend a minor revision of this work prior to publication.

Author Response

Professor Dennis Zhou

Assistant Editor of Molecules

After performing a careful and thorough review of our manuscript we managed to reply and argue about each question that the referees pointed out. We are very grateful for the high quality of the review reports which provided us with a great opportunity to considerably improve our paper. We have assigned each of the following sections to a referee, and performed the modifications according to the suggestions of the referee, as follows:

Reviewer 2

Weak Interactions, including van der Waals forces, are tremendously important and help in the development of novel molecular materials. In this work, the authors have presented a thorough computational study to understand the non-covalent interactions between molecular oxygen and noble gases. The computational data are further correlated with experimental results. In order to understand the existence of weak interactions, binding energies between noble gas-oxygen (Ng-O2) adducts were calculated. Furthermore, charge displacement analysis, symmetry-adapted perturbation theory (SAPT), and natural bond orbital (NBO) analysis were performed. Based on the computations and experiments, it was concluded that the Ng-O2 aggregates are effectively bound by van der Waals interactions, and the interactions are not affected by the charge transfer between the two. The study is interesting and will attract the attention of a broad range of audiences interested in the development of novel materials using weak forces. The work has been competently performed with a high scientific standard, and the results have been scholarly presented.

1) Referee’s comment: Though the manuscript reads well, the introduction, results, and discussion are divided into several small paragraphs which should be compiled into long paragraphs including the conclusions. Therefore, I recommend a minor revision of this work prior to publication.

Authors’ answer:  As suggested by Referee, we compiled into long paragraphs the several small paragraphs of the introduction, results and discussion, and Conclusions.